

# Observational evidence for aerosols increasing upper tropospheric humidity

Laura Riuttanen[1], Marja Bister[1], Veli-Matti Kerminen[1], Viju O. John[2,3], Anu-Maija Sundström[1,♦], Miikka Dal Maso[4], Jouni Räisänen[1], Victoria A. Sinclair[1], Risto Makkonen[1], Filippo Xausa[1], Gerrit de Leeuw[1,5] and Markku Kulmala[1]

[1]Department of Physics, University of Helsinki, P.O. Box 48, FI-00014 Helsinki, Finland

[2]Met Office Hadley Centre, FitzRoy Road, Exeter, Devon, EX1 3PB, United Kingdom

[3]EUMETSAT, Eumetsat Allee 1, D-64295 Darmstadt, Germany

[4]Tampere University of Technology, P.O. Box 692, FI-33101 Tampere, Finland

[5]Finnish Meteorological Institute, P.O. Box 503, FI-00101 Helsinki, Finland

♦ Now at EMPA, Swiss Federal Laboratories for Materials Science and Technology, Überlandstrasse 129, 8600 Dübendorf, Switzerland.

*Correspondence to*: Laura Riuttanen (laura.riuttanen@helsinki.fi)

**Abstract.** Aerosol-cloud interactions are the largest source of uncertainty in the radiative forcing of the global climate. A phenomenon not included in the estimates of the total net forcing is the potential increase in upper tropospheric humidity (UTH) by anthropogenic aerosols via changes in the microphysics of deep convection. Using remote sensing data over the ocean east of China in summer, we show that increased aerosol loads are associated with an UTH increase of $2.2 \pm 1.5$ in units of relative humidity. We show that humidification of aerosols or other meteorological covariation is very unlikely to be the cause for this result indicating relevance for the global climate. In tropical moist air such an UTH increase leads to a regional radiative effect of $0.5 \pm 0.4$ W m$^{-2}$. We conclude that the effect of aerosols on UTH should be included in future studies of anthropogenic climate change and climate sensitivity.

## 1 Introduction

The effective globally averaged radiative forcing due anthropogenic aerosols has a best estimate of -0.9 W m$^{-2}$ but a wide uncertainty range from -1.9 to -0.1 W m$^{-2}$ (Boucher et al., 2013). This estimate combines a direct forcing of -0.35 [-0.85 to +0.15] W m$^{-2}$ due to aerosol-radiation interaction with an even less well-constrained forcing due to aerosol-cloud interaction, but still omits some potentially important mechanisms of aerosol forcing. Ramanathan et al. (2001) suggested that aerosol-induced suppression of precipitation in the updrafts of deep clouds can transport more water into the upper troposphere. But





even without changes in the total amount of precipitation, changes in cloud microphysics may affect upper tropospheric humidity (UTH, Bister and Kulmala, 2011), and thereby counteract some of the cooling effects of aerosols. High values of UTH have been observed to collocate with high values of aerosol optical depth (AOD, Kottayil and Satheesan, 2015), but here we present the first observational study to show that it is very unlikely that the increase in UTH is caused by something else

than aerosols.

An increase in the number of atmospheric aerosols acting as cloud condensation nuclei (CCN) leads to more numerous and smaller cloud droplets, suppressed amount of warm rain, and enhancement of ice precipitation in deep convection (Khain et al., 2005). Sublimation of the increased amount of hydrometeors in the upper troposphere can increase the UTH (Mechanism in Fig.1), which the Earth's long-wave radiative budget is highly sensitive to (Held and Soden, 2000; Udelhofen and Hartmann,

1995). Consistent with the proposed mechanism, more extensive and longer-lasting ice anvil clouds have been observed with larger amounts of aerosols, a sign of the increased amount of hydrometeors (Koren et al., 2005; Koren et al., 2010; Fan et al., 2013). A complicating fact is that the overall effect of aerosols on climate via deep convection may be very different from their effect on single convective clouds or precipitation events. For example, if aerosols invigorate convection causing more precipitation locally, there is compensation for this effect elsewhere or at a later time because total precipitation is strongly

constrained by the atmospheric and surface energy balances (Grabowski and Morrison, 2011). Thus, a possible local increase (invigoration) or decrease (suppression) in the amount of precipitation caused by aerosols, has to be accounted for in the studies of the effect of aerosol-cloud interactions on climate (Invigoration problem in Fig. 1), as we do in this study. Note that in the following, we use the term 'invigoration' for both positive and negative changes in precipitation caused by aerosols.

In this work, we investigated the effect of aerosols on UTH by using satellite data of UTH and AOD in a study region east of

China. We also used global data to conduct a preliminary study of the effect elsewhere. To exclude the invigoration effect from our analysis (see Fig. 1), we binned our data according to the amount of precipitation and compared UTH in cases of high and low AOD but with similar amount of precipitation. We show, for the first time, observational evidence that strongly suggests that aerosols increase the UTH.

## 2. Data and methods

### 2.1 Data

UTH, i.e. the relative humidity with respect to water, in a layer between approximately 200 and 500 hPa, was obtained from the microwave humidity sounder (MHS; Bonsignori, 2007) onboard the MetOp-A satellite for 60° S to 60° N. The ascending node was used, crossing the equator northward at 9:30 p.m. local solar time. A microwave method developed by Buehler *et al.* (2008) enabled us to detect relative humidities also in the areas of anvil clouds.

The daily rainfall product, 3B42, from the Tropical Rainfall Measuring Mission (TRMM) multisatellite analysis (Huffman et al., 2007) was used in this study. These data cover the globe from 50° S to 50° N and are provided on a 0.25° × 0.25° grid. The



TRMM daily rainfall product consists of satellite and in situ measurements during a 24 h time period starting at 22:30 UTC of the previous day (in our study region approximately 8:30 a.m. local solar time).

AOD has in general been considered as a moderate proxy of cloud condensation nuclei (CCN; Andreae, 2009). We used AOD from the Moderate Resolution Imaging Spectroradiometer (MODIS) instrument onboard the Terra satellite that crosses the equator southward at 10:30 a.m. local solar time (Remer et al., 2005). MOD08_D3 collection 5 level 3 global daily 1° × 1° data "Effective Optical Depth Average Ocean Mean" at 550 nm was used in most parts of the study, and "Optical Depth Land and Ocean Mean" in global analysis. MODIS level 3 products are obtained from level 2 swaths along the flight, with 10 km resolution for AOD and uncertainty of ±0.03±0.05×AOD over ocean. A satellite instrument cannot retrieve AOD below clouds, and in level 2 data also the pixels adjacent to a cloud are removed from the AOD product to avoid possible cloud contamination of the data. However, as the 1° × 1° level 3 data is produced from the 10 km-resolution level 2 data, it is possible to obtain AOD and cloud properties for the same 1° × 1° grid cell when this cell is partly cloudy (Fig. 2).

## 2.2 Cirrus correction

Cirrus clouds may contaminate the AOD values obtained from MODIS (Sun et al., 2011; Wollner et al., 2014; Huang et al., 2011). In our study region, MODIS AOD increased almost linearly with the increasing cirrus fraction (cf), obtained also from MOD08_D3 (Figs. 3 and 4). This may partly be due to a microphysical effect of aerosols, as more extensive cirrus anvils have been observed in association with higher AOD (Koren et al., 2005; Koren et al., 2010; Fan et al., 2013 and as shown schematically in Fig. 1a and b). However, AOD data contamination due to sub-visual cirrus is also possible (Sun et al., 2011; Wollner et al., 2014; Huang et al., 2011).

We removed data with cirrus fraction larger than 0.9 and used linear regression for cirrus fractions between 0.1 and 0.9 to remove the cirrus dependence from the AOD data to minimize the possible AOD data contamination (as explained in Fig. 4). However, by doing so, we also removed any possible microphysical effect of aerosols on cirrus anvils and subsequently may have underestimated the aerosol effect on UTH. Therefore, we also report results without the cirrus correction in Table 1. It should be noted that the applied cirrus correction is designed to only assure that our results are not contaminated by sub-visual cirrus; it is not therefore a general method to be applied in all satellite data.

For sensitivity tests (Table 2), a linear regression for the cirrus fractions between 0.1 and 0.9 was applied with a fit of AOD = $0.1486 \times cf + 0.2535$ and root mean squared error of 0.3690 (cirrus correction version 2). The grid cells with a cirrus fraction larger than 0.9 were excluded from the analysis. For global data a similar cirrus correction as explained in Fig. 4 was applied, with a linear fit to global data of all seasons 2007-2013: AOD = $0.072 \times cf + 0.100$. The cirrus correction may not be geographically or seasonally homogeneous, but for simplicity we applied the same correction to all grid cells in the global analysis.





### 2.3 Study period and area

We investigated the effect of aerosols on UTH in the maritime region east of China (120-149° N, 25-45° E). This area was chosen because of the advection of anthropogenic aerosols from the continent to this region and the frequent occurrence of convection (Fig. 5). We analysed data from days with precipitation for June, July and August in 2007-2013. Only data from the summer period were considered because during the other seasons, the advective wind speeds are much higher so that the 11-hour time difference between the AOD and UTH measurements could have resulted in these two quantities being determined from totally different air masses.

We used measurements of AOD preceding the measurements of UTH by 11 hours to better allow convection and microphysical changes, including sublimation of ice, to occur between the AOD and UTH measurements. Eleven hours may not be long enough to allow the sublimation of all ice associated with the large ice anvils produced in deep convection but a longer time lag could potentially lead to problems due to the horizontal advection of air. AOD was obtained from the Terra satellite crossing the equator southward at 10:30 a.m. local solar time and UTH from the MetOp-A satellite crossing the equator northward at 9:30 p.m. local solar time.

### 2.4 Data analysis

All the data were linearly interpolated to the same $1° \times 1°$ grid. The UTH data were binned according to daily precipitation in the same $1° \times 1°$ grid. The applied data were restricted to those cases when the daily precipitation was more than 1 mm. For each grid point, $\Delta$UTH was calculated as $\Delta$UTH = median{UTH[AOD≥median(AOD)]} – median{UTH[AOD<median(AOD)]} for each precipitation bin, and an average was then calculated over the precipitation bins. Median(AOD) is the median of all AOD data in the study region for precipitation values of more than 1 mm day$^{-1}$. For global analysis median(AOD) is the median of AOD in each grid box. The 90% confidence interval of the mean was calculated as $\pm 1.725 \times SD/\sqrt{N}$, where SD is the standard deviation of the N = 21 monthly median values, which were assumed to be independent of each other. For global analysis, a two-sided t-test was conducted over the monthly medians and only values significant at the 10% level are shown. A logarithmic fit was calculated for the binned mean values in Fig. 6a, excluding the first data point. The obtained fit is UTH = 45.5015 + 1.3680 × ln(AOD).

### 2.5 Radiative transfer calculations

The sensitivity of the radiative transfer to perturbations in the UTH was tested by using the libRadtran radiative transfer code (Mayer and Kylling, 2005). Reference soundings for the tropical atmosphere (Dunion, 2011) were used. We added 1, 2, 5 and 10 %RH to the 500-200 hPa layer of the reference soundings of tropical moist (TM) and mid-latitude dry (MLD) air. The outgoing long-wave flux at the top of the atmosphere was integrated over the 4-50 micron wavelength band using the DISORT radiative transfer equation solver (Stamnes et al., 1988) and the correlated k-method by Fu and Liou (1992). The radiative



effects were calculated only for the long-wave region, as the shortwave effect was estimated to be minor. Stratospheric adjustment was not taken into account.

## 3. Results

### 3.1 UTH increase associated with AOD

We found that UTH increased with increasing values of AOD for AOD higher than about 0.08 (Fig. 6a). From a logarithmic fit to these values, the UTH increase associated with the AOD increase, for example from 0.1 to 0.6, was observed to be 2.5 %RH. An UTH increase was not visible with very high amounts of precipitation (Fig. 7), presumably due to enhanced wet scavenging of aerosols. Namely, earlier strong precipitation may result in larger UTH occurring together with smaller AOD, which then masks the microphysical effect of aerosols on UTH (Quaas et al., 2010; Grandey et al, 2013). Overall, wet scavenging associated with both earlier and simultaneous precipitation masks the effect of aerosols on UTH and therefore our results may underestimate the true effect.

The median AOD over the study region was 0.18 for precipitation values of more than 1 mm day$^{-1}$. We selected this median to represent the limit between "high" and "low" AOD. The mean UTH difference between these two AOD categories (ΔUTH) was 2.2 ± 1.5 %RH (90% confidence interval, see Methods) for precipitation values of more than 1 mm day$^{-1}$ (Fig. 6b, see also Fig. 8). Without applying the cirrus correction, which removes part of the real microphysical effect, the corresponding UTH difference was much higher, 5.8 ± 1.4 %RH (Table 1). Without the precipitation binning, or with only a few bins, a larger mean difference in UTH between cases with "high" and "low" AOD was obtained (Fig. 9). UTH also increased less as a function of AOD in Fig. 6a when averaged over 10 precipitation bins than when the median values without precipitation binning were used. The fact that precipitation binning decreases the obtained ΔUTH may be at least partly due to the removal of invigoration effects of aerosols on precipitation.

We tested the sensitivity of the results by varying the limit between "high" and "low" AOD and also by averaging AOD over 5° × 5°. We also varied the time lag between AOD and UTH measurements, and the time span over which the precipitation was taken, up to 2 days. Furthermore, different aerosol products were tested. In general, a higher UTH was observed with a higher value of AOD. However, the magnitude of ΔUTH depended on the specific choices (Table 2).

### 3.2 Causality questions

Causalities in satellite studies of aerosol-cloud-interactions have been questioned because of meteorological covariation and aerosol humidification (Quaas et al., 2010; Grandey et al., 2013). For example, aerosol particles have been noted to swell in humid conditions leading to higher AOD even though the number of cloud condensation nuclei does not change (e.g. Quaas et al., 2010; Grandey et al., 2013; Twohy et al, 2009). Such effects may then cause correlations between AOD and cloud properties.





Although precipitation binning should alleviate these problems, we repeated our analysis by binning the data according to different meteorological variables from ERA-Interim reanalysis data (Dee et al., 2011), in addition to binning it by the amount of precipitation. Surface parameters and 550 hPa vertical velocity in pressure coordinates at 00 UTC (approximately 10 a.m. local solar time) were used at $1° × 1°$ resolution (Fig. 10).

If meteorological covariation or humidification explained our results, binning by meteorological variables should significantly decrease the observed UTH difference between the high and low values of AOD. For cases with non-extreme values for meteorological variables (i.e. boxes 2 to 5 in Fig. 10), the results were practically the same as with binning only by precipitation. With extreme values of some meteorological parameters (the lowest MSL pressure, the highest 2-meter relative humidity and the largest pressure increase from the previous day), the results showed a weaker or even a negative relation between AOD

and UTH.  However, those values were likely associated with current or recent passage of low pressure systems, where the wet scavenging associated with significant amount of earlier precipitation may dominate the results.  We conclude that meteorological covariation and humidification were not the reason for the observed association between UTH and AOD. This strongly suggests a causal relationship between aerosols and UTH.

### 3.3 Effect on radiative transfer

Outgoing long-wave radiation is sensitive to the upper tropospheric water vapour concentration (Held and Soden, 2000). According to our radiative transfer calculations (see Section 2.5), a 1 %RH increase in the UTH of moist tropical air causes a local positive radiative effect of 0.25 W m$^{-2}$ (0.39 W m$^{-2}$ for dry air, see Table 3). The observed summertime increase of 2.2 ± 1.5 %RH in UTH (5.8 ± 1.4 %RH without removal of AOD dependence on cirrus fraction) due to aerosols corresponds to the radiative effect of +0.5 ± 0.4 W m$^{-2}$ (+1.4 ± 0.3 W m$^{-2}$, interpolated from Table 3). The real radiative effect by this phenomenon

may be even larger than these values, since the effect of wet scavenging may partially obscure the microphysical effects of aerosols on UTH in our study region.

### 4. Discussion and conclusions

Although China's outflow region is an optimal study region because of the aforementioned reasons, it is important to consider the relation between UTH and AOD globally. In the following we make a first attempt to understand the effect on global

scales, but a more in-depth analysis would be a topic of an additional study. In Fig. 11, differences in UTH between the cases of "high" and "low" AOD averaged over three months are shown for the four seasons separately. The UTH difference was mostly positive, which supports our hypothesis, even though negative values were also observed in some locations, especially over remote oceans and in parts of the intertropical convergence zone. These negative values are mostly located far away from major aerosol sources and may thus be due to wet scavenging dominating the relationship between AOD and UTH over the

effect of aerosols on the amount of ice in the upper troposphere and its sublimation. Similarly, where the ΔUTH is positive, it would be even more positive without wet scavenging. As wet scavenging overall leads to a smaller ΔUTH, it also decreases



the statistical significance of the obtained results. Importantly, Fig. 11 indicates that our study region east of China is not anomalous, but that similar differences in UTH between "high" and "low" AOD are observed in many other regions over the globe.

A few caveats need to be taken into account when interpreting the global results (Fig. 11). First of all, in our study area, binning
by other meteorological parameters proved unnecessary when the data were first binned by the amount of precipitation. However, in theory the same might not apply globally. Also, the AOD dependence on cirrus fraction may not be the same everywhere, as has been assumed in these calculations. However, changes in cloud microphysical properties tend to show logarithmic, rather than linear, dependence on the aerosol loading (Koren et al., 2008 and references therein), suggesting that the effect may also be large in cleaner areas.

Results from general circulation models show almost no changes in the relative humidity in a changing climate (Held and Soden, 2000; Soden et al., 2005). However, very few general circulation models have so far begun to include effects of aerosols on deep convective clouds (Boucher et al., 2013; Khain et al., 2015). Namely, as noted by Khain et al. (2015), the only feasible option in current general circulation models is to use bulk microphysics parameterization schemes. However, it has been shown that bulk microphysics schemes cannot capture important aerosol effects on deep convection. For example,
increased cloud top height and larger anvil clouds seen in observations, are not produced by bulk schemes. Only spectral bin microphysics, which is computationally too costly and not an option for current general circulation models, can produce such effects of aerosols (Khain et al., 2015). Since existing general circulation models cannot simulate aerosol effect on UTH, this phenomenon is not included in the current projections of climate change. Estimates of climate sensitivity that combine temperature observations with estimates of greenhouse gas and aerosol induced radiative forcing should also be reconsidered.

**Author Contributions**

L.R. and M.B. designed the study and wrote the paper. L.R. conducted the study. V.-M.K. participated in writing of the paper. V.J. processed the UTH data. A.-M.S. conducted the LibRadTran runs. All authors discussed the methods and results and commented on the manuscript.

**Acknowledgments**

This work was supported by the Academy of Finland Center of Excellence program (project number 272041). MB would like to acknowledge the contribution of the COST Action ES0905. VOJ was supported by the U.K. Department of Energy and Climate Change (DECC) and Department of Environment, Food and Rural Affairs (DEFRA) Integrated Climate Programme (GA01101) and the EUMETSAT CMSAF. The authors want to thank ECMWF for producing the ERA-Interim dataset as well as NASA MODIS Atmosphere and TRMM teams for providing data at their data servers. We thank Stephanie Lindström for
producing Fig. 1. We also want to thank Prof. Heikki Järvinen for reading the manuscript and giving helpful comments.



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





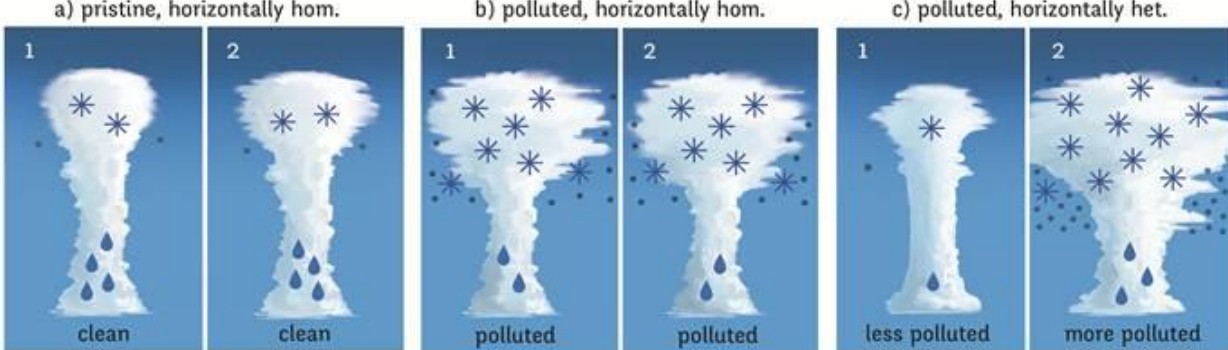

**Figure 1: Deep convection in a) pristine, horizontally homogeneous, b) polluted, horizontally homogeneous and c) polluted, horizontally heterogeneous conditions. In all pairs of figures (a, b and c) the total amount of precipitation reaching the ground is the same. i) *Mechanism*. Aerosols decrease warm rain, increase ice precipitation and increase hydrometeor mass in the upper troposphere as shown in (a) and (b). More sublimation in (b) than in (a) causes an increase in the UTH in (b). ii) *Invigoration problem*. Horizontal variation of the aerosol amount, as within (c), can invigorate convection in more polluted areas (box 2 in Fig. c) and suppress convection in cleaner areas (box 1 in Fig. c). Therefore, comparing the conditions of box 2 in (c) to that in (a) would make the effects of aerosols seem too large. We avoid this problem by comparing cases with similar amounts of precipitation.**





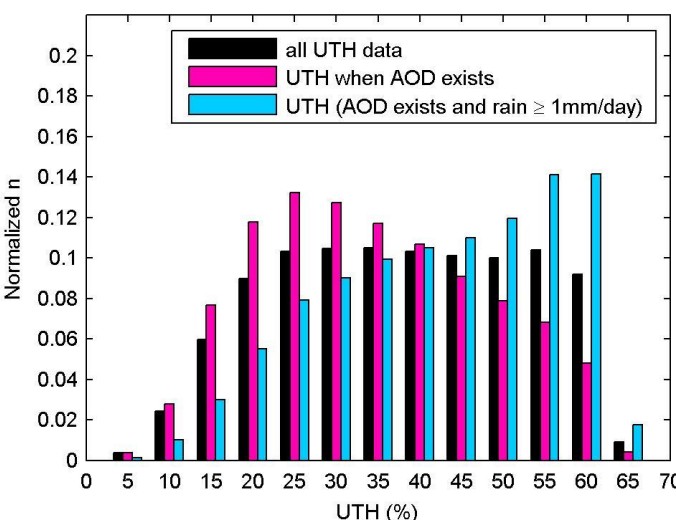

**Figure 2: Number of data.** Histograms of the UTH data east of China in June, July and August in 2007 - 2013. AOD cannot be retrieved in the presence of clouds, which means that we were lacking part of the data with the highest values of UTH. AOD was retrieved in approximately one third of the grid cells, in 106580 data points of the total 350407. When we limited our study to cases with precipitation ≥ 1mm day⁻¹, we had 17318 data points and the relative number of low UTH values decreased. UTH is strongly linked to precipitation, as both depend on the activity of convection.





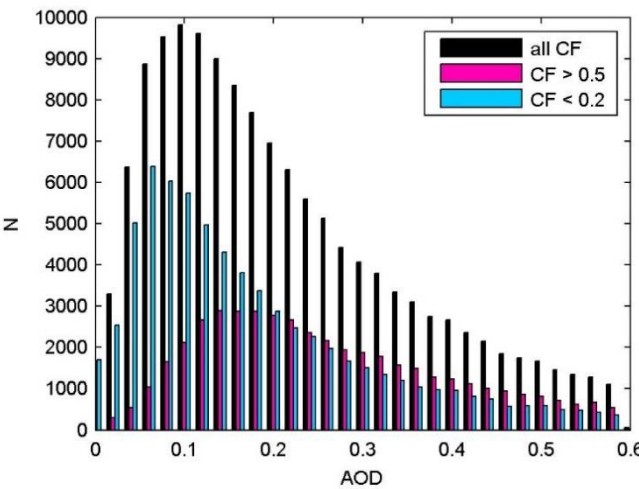

**Figure 3: The relation between cirrus fraction and AOD. Histogram of MODIS AOD for all the data and segregated for small (<0.2) or large (>0.5) cirrus fraction (CF) in the same 1° × 1° grid east of China in June, July and August in 2007 - 2013. For visual clarity, the AOD data considered here are restricted to values smaller than 0.6. The difference between these two distributions indicates that cirrus may contaminate the AOD data retrieval. Therefore, we removed the cirrus dependence from the AOD data (as explained in Fig. 4) to ensure that the relationship between AOD and UTH is not due to data contamination.**





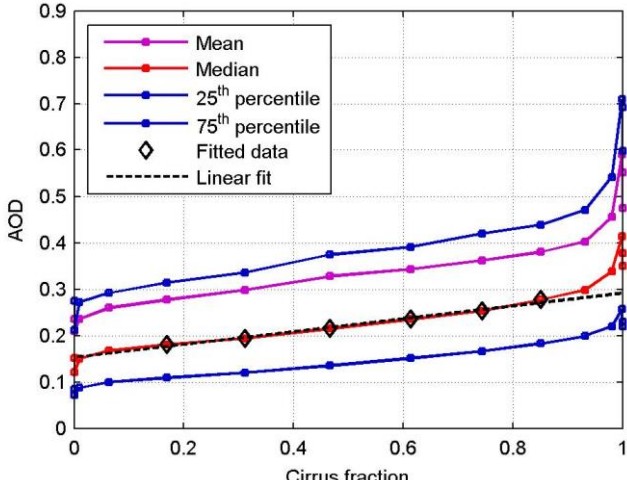

**Figure 4: Dependence of AOD on the cirrus fraction. MODIS AOD at 550 nm as a function of cirrus fraction (cf) in the same 1° × 1° grid east of China in June, July and August in 2007 - 2013. The dashed line shows a linear fit (AOD = 0.139 × cf + 0.153) obtained using the median data. In order to eliminate any possible AOD data contamination by cirrus, all AOD values for the cirrus fraction between 0.1 and 0.9 were adjusted (AOD') based on the following linear fit: AOD' = AOD − 0.139 × cf + 0.014. No correction was applied to the data for the cirrus fraction smaller than 0.1, and the data with cirrus fraction larger than 0.9 were not used in this study.**





**Figure 5: The study region and the data used. (a) AOD, (b) daily precipitation, (c) 350 hPa wind roses and (d) UTH in the study region east of China averaged from the observations in June, July and August in 2007 - 2013. a, Aerosol outflow from continental China clearly dominates AOD in the region, with contributions from Japan and Korea. b, Precipitation is plentiful in the region. c, Mean wind direction in the region is westerly. d, There are high UTH values downwind of the regions with large values of precipitation.**





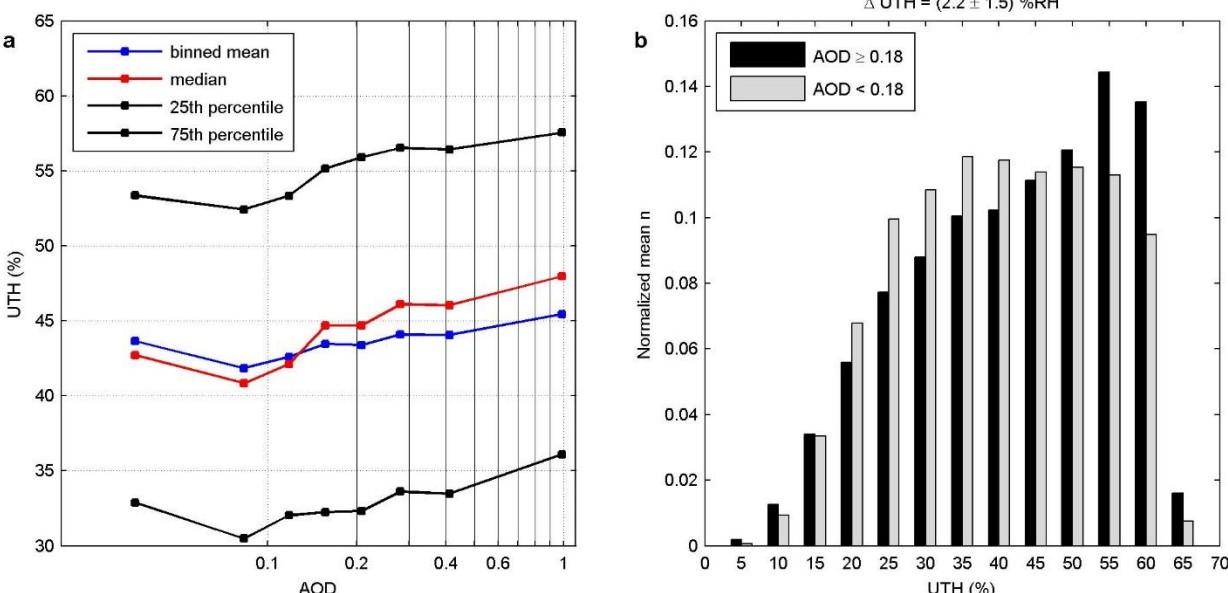

**Figure 6: a, UTH as a function of AOD and b, histogram of UTH in case of low and high AOD east of China (120-149° E, 25-45° N) in June, July and August in 2007 - 2013. In both figures only data from 1° × 1° grid cells with precipitation values of more than 1 mm / day are shown. Binned mean values are averages over 10 precipitation bins. Cirrus dependence has been removed from the AOD data (see Fig. 4).**



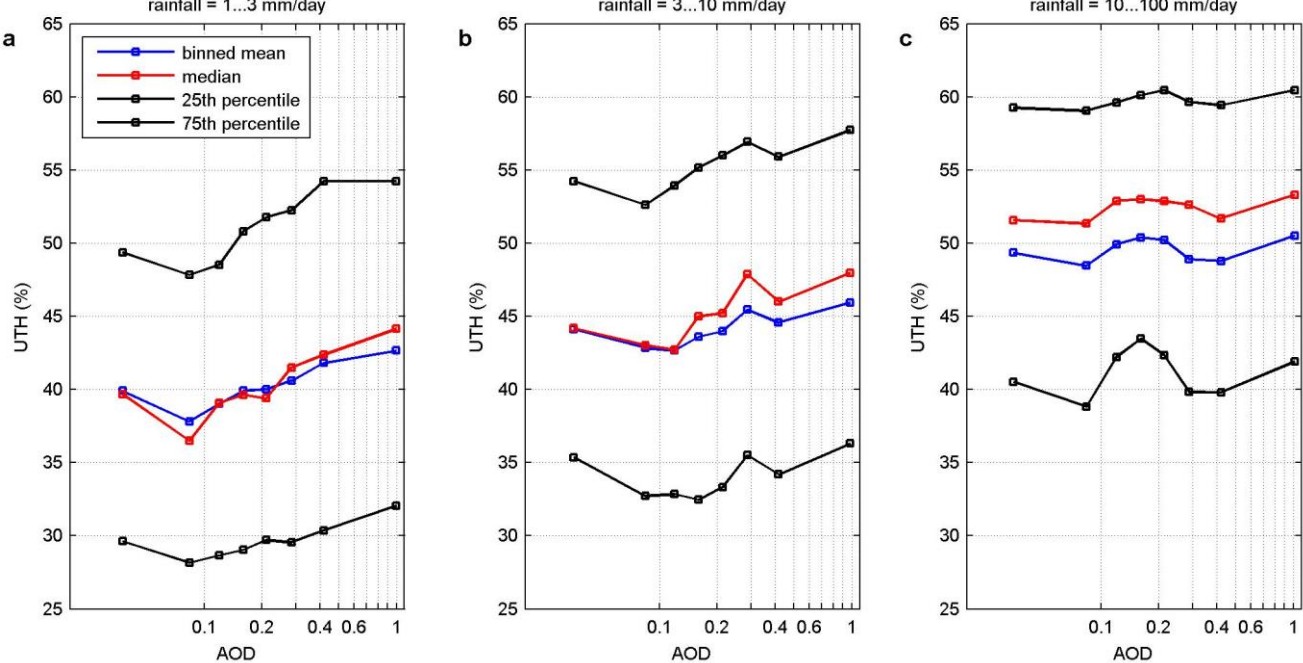

**Figure 7: Results in three precipitation categories. UTH as a function of AOD (on a logarithmic scale) in three precipitation categories: (a) 1-3 mm day$^{-1}$, (b) 3-10 mm day$^{-1}$ and (c) 10-100 mm day$^{-1}$, east of China in June, July and August in 2007 - 2013. In each precipitation category, the AOD data were split into 8 bins, after which the 25th, 50th and 75th percentiles of the data and mean UTH of 10 precipitation bins were calculated. The cirrus dependence has been removed from the AOD data.**




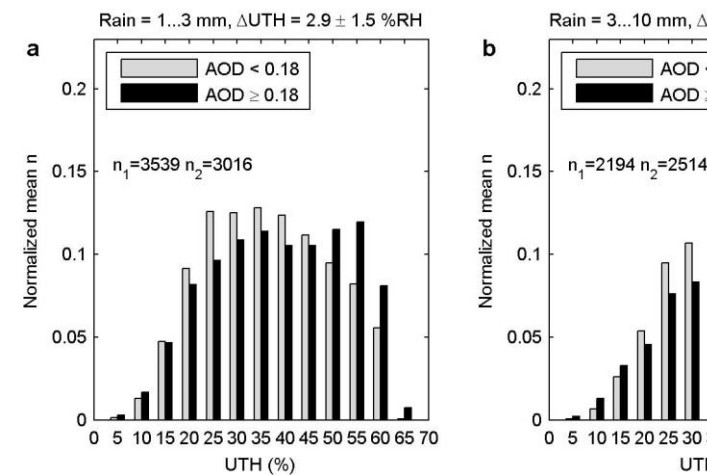
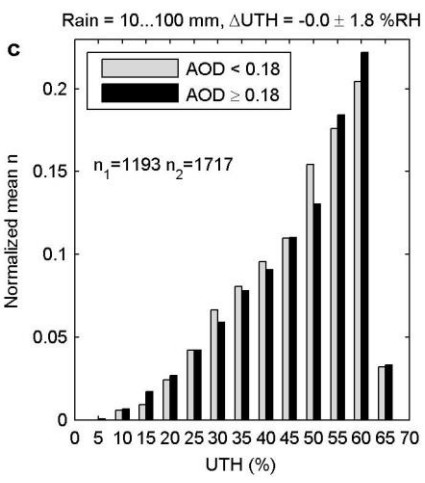

**Figure 8: UTH differences in three precipitation categories.** Histograms of UTH in case of "low" and "high" AOD east of China in June, July and August in 2007 - 2013, for three categories of daily precipitation: (a) 1-3 mm, (b) 3-10 mm and (c) 10-100 mm. The y-axis shows the normalized mean histogram of UTH, which is the mean of 10 precipitation bins in each precipitation category. ΔUTH was calculated as mean of {median[UTH(AOD≥0.18)]-median[UTH(AOD<0.18)]}, while its 90% confidence interval was calculated from the standard error of the monthly mean. The values are presented as percentage units. n1 and n2 show the number of data points in "low" and "high" AOD cases, respectively. The cirrus dependence has been removed from the AOD data.





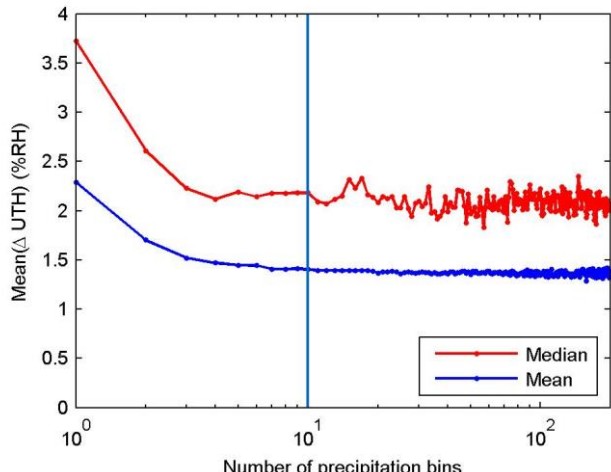

**Figure 9: Effect of precipitation binning. Change in UTH (ΔUTH = UTH(AOD≥0.18) - UTH(AOD<0.18)) as a function of the number of precipitation bins over which the average is calculated for the data east of China in June, July and August in 2007 - 2013. Here, only the precipitation values of more than 1 mm day$^{-1}$ were taken into account. The cirrus dependence has been removed from the AOD data. When no precipitation binning was applied, or when only a few bins were used, the increase in UTH was clearly larger than when at least 10 bins were used. This indicates that the precipitation binning is essential if we want to remove the effect of convective invigoration on UTH. In this study, we used 10 precipitation bins (indicated by the vertical line in the figure) in order to remove the influence of invigoration on our results.**



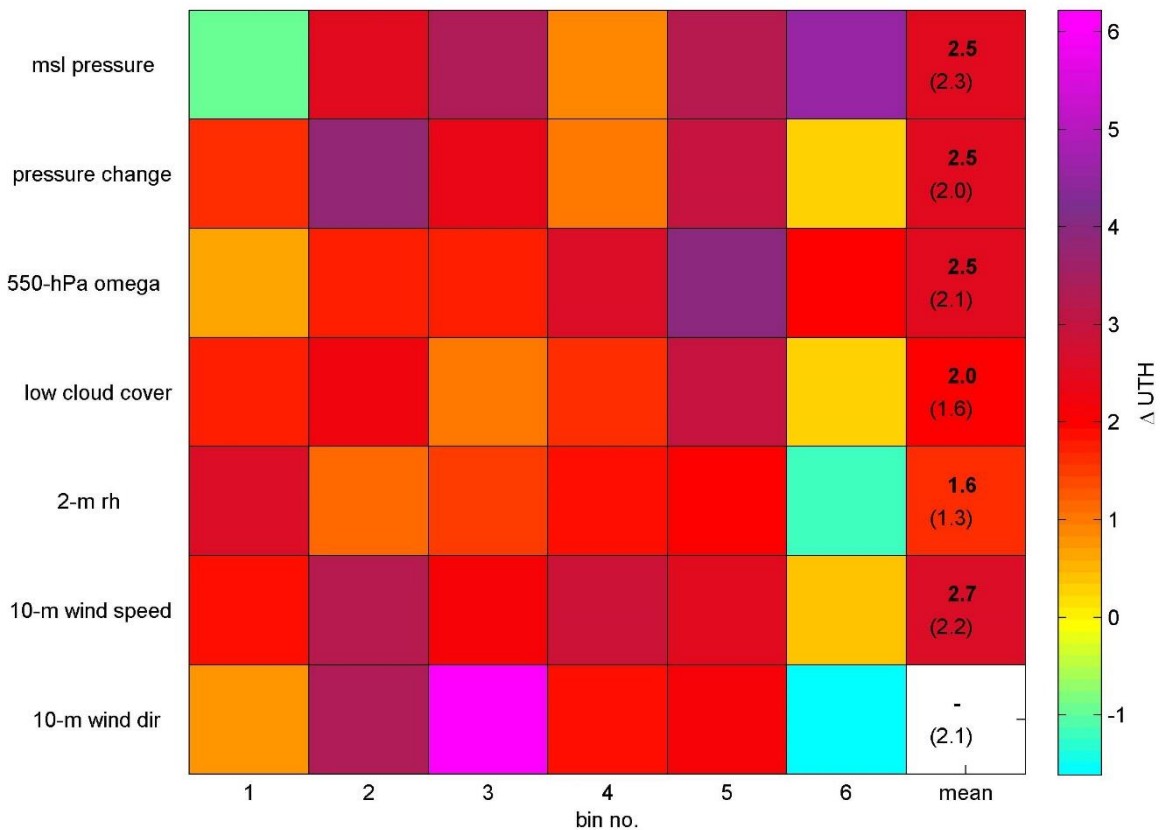

**Figure 10: Meteorological covariation.** After binning with precipitation, data was binned to six bins according to the value of several meteorological variables from ERA-Interim reanalysis: mean sea level pressure, pressure change from the previous day, 550-hPa vertical velocity in pressure coordinates, low cloud cover, 2-m relative humidity, 10-m wind speed and 10-m wind direction. These variables were chosen as they are expected to change considerably during the passage of synoptic (and mesoscale) disturbances. Change in UTH ($\Delta$UTH = UTH(AOD$\geq$0.18) - UTH(AOD<0.18)) is shown separately for the six bins and the mean over the six bins in the last column. For each meteorological variable, each of the six bins contains equal amount of data. The values increase from left to right except for 10-m wind direction for which the values go from north-easterly to north-westerly wind directions. In the last column, the upper value is the mean of bins 2-5 and the mean of all six bins is shown in parenthesis.

If the observed increase in UTH would be caused by some of these meteorological variables, $\Delta$UTH should go to zero when binned according to this variable. As no such behaviour is observed, we conclude that meteorological covariation is not the cause of the obtained results. Also, $\Delta$UTH shows no systematic behaviour according to any of the meteorological variables.



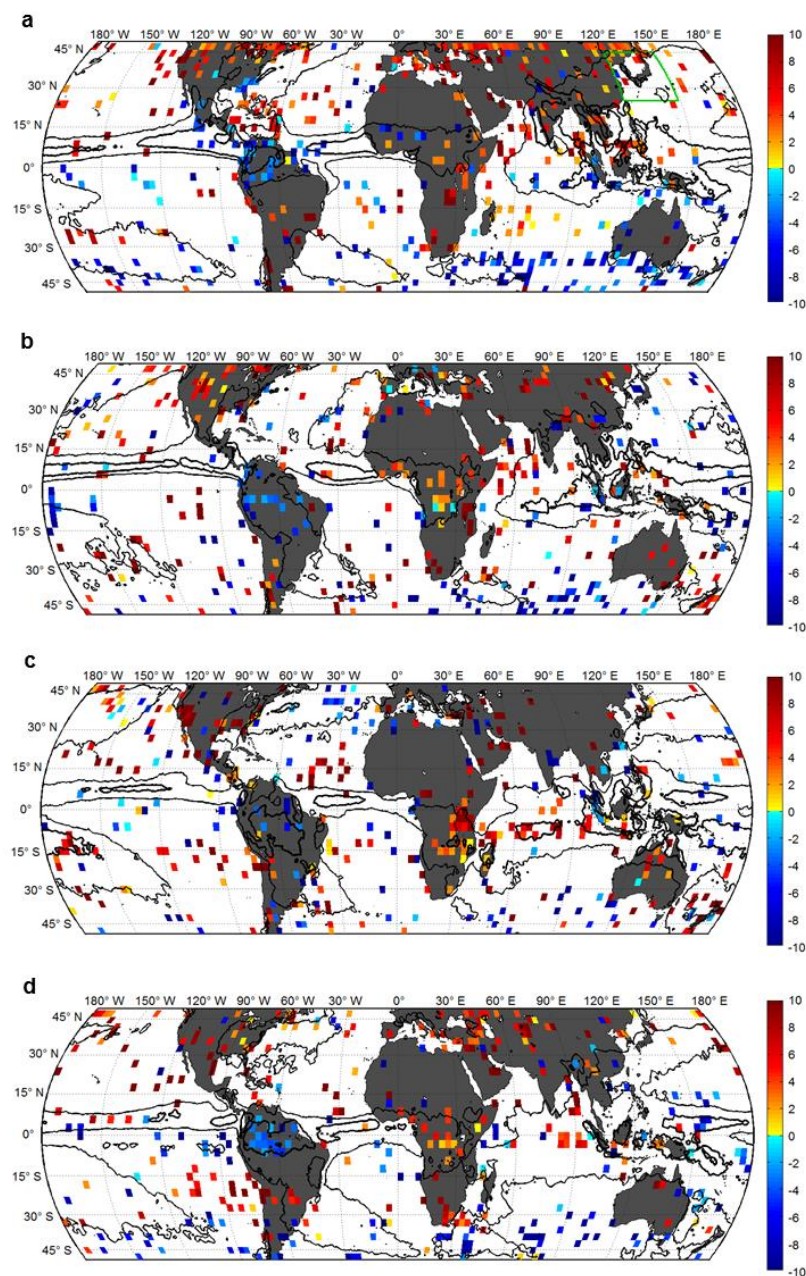

**Figure 11: Mean difference in UTH (in %RH) between cases when AOD is higher and cases when AOD is lower than each grid box's median for four seasons in 2007 – 2013: (a), June, July, August, (a) September, October and November, (b) December, January and February, (c) March, April and May. Precipitation binning and cirrus correction were applied to the data and only grid cells with precipitation more than 1 mm / day were used. The data are averaged over 3° × 3° grid and only grid cells with statistically significant differences (t-test with the 10% significance level) are shown. Black contours represent mean daily precipitation of 3 mm, 9 mm and 30 mm. The study region east of China is marked with a green box in (a). Wet scavenging is likely to decrease ΔUTH leading also to smaller statistical significance.**





**Table 1: Impact of different choices on the results. ΔUTH was calculated as ΔUTH=UTH[AOD≥median(AOD)]-UTH[AOD<median(AOD)] east of China in June, July and August in 2007 - 2013 and is expressed as %-units. 90% confidence interval is calculated from the standard error of the monthly mean ΔUTH.**

| | No cirrus correction | Dependence on cirrus fraction removed from the AOD data |
|---|---|---|
| ΔUTH | 5.6 ± 0.9 | 1.9 ± 0.8 |
| ΔUTH, for precipitation > 1 mm day$^{-1}$ | 7.3 ± 1.7 | 3.6 ± 1.8 |
| ΔUTH, for precipitation > 1 mm day$^{-1}$, precipitation binning applied with 10 precipitation bins | 5.8 ± 1.4 | 2.2 ± 1.5 |



**Table 2: Results from the sensitivity tests. ΔUTH is calculated for different MODIS aerosol products, for UTH from different overpassing times and satellites, and for different precipitation measurement time spans. Small ΔUTH values are obtained when there is precipitation on the previous day (i.e. wet scavenging) or when UTH is measured before AOD (causality).**

| AOD | UTH | precipitation | ΔUTH | err | note |
|---|---|---|---|---|---|
| $AOD_{550}$ | MetOp-A$_{asc}$ | TRMM daily | 2.16 | 1.50 | |
| $AOD_{550, 0.2}$ | MetOp-A$_{asc}$ | TRMM daily | 2.30 | 1.28 | AOD limit 0.20 |
| $AOD_{550, ci2}$ | MetOp-A$_{asc}$ | TRMM daily | 2.00 | 1.45 | cirrus correction version 2 |
| $AOD_{land\&ocean}$ | MetOp-A$_{asc}$ | TRMM daily | 2.02 | 1.05 | same cirrus correction as for AOD 550 |
| AI | MetOp-A$_{asc}$ | TRMM daily | 1.94 | 1.59 | AI = AOD 550 * Ångström exponent |
| $AOD_{550}$ | MetOp-A$_{desc}$ | TRMM daily | 0.77 | 0.81 | UTH measured 1h before AOD |
| $AOD_{550}$ | NOAA-18$_{asc}$ | TRMM daily | 1.18 | 1.13 | UTH measured 9h before AOD |
| $AOD_{550}$ | NOAA-18$_{desc}$ | TRMM daily | 2.42 | 1.26 | UTH measured 3h after AOD |
| $AOD_{550}$ | MetOp-A$_{asc}$ | TRMM daily + previous day | 0.75 | 0.64 | |
| $AOD_{550}$ | MetOp-A$_{asc}$ | TRMM daily + following day | 1.99 | 0.81 | |



**Table 3: Radiative effect related to ΔUTH. Outgoing longwave radiation (OLR) and radiative effect (RE) caused by a decrease in OLR when relative humidity in the upper troposphere (500-200 hPa) is increased by 1, 2, 5, and 10 percentage units in the tropical moist (TM) and mid-latitude dry (MLD) air reference soundings of Dunion (2011).**

|  | TM | | MLD | |
|---|---|---|---|---|
|  | OLR (W m$^{-2}$) | RE (W m$^{-2}$) | OLR (W m$^{-2}$) | RE (W m$^{-2}$) |
| Dunion (2011) | 256.83 | 0 | 269.07 | 0 |
| +1 %RH | 256.58 | 0.25 | 268.68 | 0.39 |
| +2 %RH | 256.33 | 0.50 | 268.30 | 0.77 |
| +5 %RH | 255.61 | 1.22 | 267.21 | 1.86 |
| +10 %RH | 254.46 | 2.37 | 265.55 | 3.52 |

