# Peer review of "Observational evidence for aerosols increasing upper tropospheric humidity"

_Atmospheric Chemistry and Physics, 2016_

## Referee Comment (RC1) · Anonymous Referee #1 · 19 Aug 2016

General Comments:

This is a very interesting study to investigate the impacts of aerosol on the increase in upper tropospheric humidity by using remote sensing datasets over the ocean east of China. The study shows that increased aerosol loads are associated with higher upper tropospheric humidity via changes in the microphysics of deep convection. Based on long-wave radiation transfer calculation, the authors concluded that an increase in upper tropospheric humidity leads to a positive regional radiative effect. The results are well presented and structured, and the topic is suitable for publication in Atmos. Chem. Phys. after addressing some specific comments listed below.

Specific Comments:

An increase in the number of atmospheric aerosols acting as cloud condensation nuclei (CCN) would slow down the diffusion growth of droplets, and thus smaller cloud droplets. To better support this mechanism using observational evidences, cloud particle effective radii (or cloud albedo) and cloud fraction, which can be retrieved by remote sensing observations, are suggested to be included in the analysis.

Identification of deep convective clouds might further improve the results. MODIS-detected cloud top pressure and CloudSat data would be helpful to select the cases with deep convective clouds.

It should be noted and be mentioned in the article that using AOD as a measure of CCN concentration may introduce substantial uncertainties which is dependent on the aerosol type, vertical profile and hygroscopic growth.

The discussion on long-wave radiative effect is somewhat incomplete and unclear. For instance, the time period of the calculated the top-of-the-atmosphere radiative effect should be clarified, monthly mean or instantons values, in Section 2.5. Moreover, discussions on radiative effect are too short in Section 3.3. It would be better to discuss the radiative effect in detail, such as the difference between tropical moist (TM) and mid-latitude dry (MLD) conditions.

Technical Corrections:

Caption of Figure 11: please check the labels of panels in the caption. Page 6, Line 8: MSL should be defined here. Figure 10: the abbreviations like omega and dir in this figure need to be specified in the caption.

---

## Referee Comment (RC2) · Anonymous Referee #2 · 1 Sep 2016

This is an interesting paper that studied the relationships between aerosol loading and upper tropospheric humidity using satellite remote sensing data. The paper is very well written and easy to follow. The authors have comprehensively explored many elements of uncertainty, however I still have some questions on the analysis. I suggest the publication of the manuscript after addressing some comments as follows.

Specific comments:

1. During the summer season, there might be diurnal variation of precipitations. Since the UHR change could result from the convective transport of marine boundary layer air to the upper troposphere and AODs are measured 11 hours ahead of URH, differences in diurnal variability between precipitation (UHR) and AODs are likely to contribute to the seen relationships. Suggest test it.

[Figure]

2. The binning approach is not totally convincing for excluding the meteorological/synoptic effects on AOD. In addition, the value ranges of those bins are not clear. I would think more logically one should compare the 10-m wind speed and surface humidity between high AOD and low AOD scenarios and test the statistical significance of 10-m wind speed differences and surface humidity differences between the high and low AOD scenarios. Not only that AODs are influenced by humidity but also it has significant contributions from large size natural sea salt aerosols (their emission rates are wind speed dependent).

3. Is it possible to only use clear sky AOD (with cirrus fraction = 0 or < 0.1), given that cirrus clouds lead to biases in AOD values? Does the results change when using only clear sky AODs?

Minor comments:

Page 7, Lines 10 and 15: I disagree that bulk microphysics cannot capture important aerosol effects, and only spectral bin microphysics can capture the effect. Through improved coupling between aerosols and microphysics in conjunction with maybe some improvement in the bulk microphysics (including ice nucleation scheme), there is no reason that more detailed bulk microphysics cannot simulate those effects.

---

## Author Comment (AC1) · 14 Oct 2016

Observational evidence for aerosols increasing upper tropospheric humidity

Reply to Anonymous Referee #1

Laura Riuttanen et al.

Atmospheric Chemistry and Physics Discussions 14.10.2016

Comments by Referee #1:

**General Comments:**
**This is a very interesting study to investigate the impacts of aerosol on the increase in upper tropospheric humidity by using remote sensing datasets over the ocean east of China. The study shows that increased aerosol loads are associated with higher upper tropospheric humidity via changes in the microphysics of deep convection. Based on long-wave radiation transfer calculation, the authors concluded that an increase in upper tropospheric humidity leads to a positive regional radiative effect. The results are well presented and structured, and the topic is suitable for publication in Atmos. Chem. Phys. after addressing some specific comments listed below.**

Authors' response:

We thank the Anonymous Referee #1 for his / her comments. Please find our response to the specific comments below.

**Specific Comments:**
**An increase in the number of atmospheric aerosols acting as cloud condensation nuclei (CCN) would slow down the diffusion growth of droplets, and thus smaller cloud droplets. To better support this mechanism using observational evidences, cloud particle effective radii (or cloud albedo) and cloud fraction, which can be retrieved by remote sensing observations, are suggested to be included in the analysis.**

The referee suggests adding cloud properties by satellite instruments to the study. Cloud processes are an essential step in the proposed AOD-UTH causality chain. However, as deep convective clouds develop quite rapidly, data with a very fine time resolution would be needed. We definitely think such a study would be worthwhile, but would not be within the scope of this paper as a different source of data would be needed (e.g. from a measurement campaign).

Several studies have already shown that in polluted environments cloud droplets and ice crystals associated with deep convective clouds are smaller and more numerous than in clean environments. Both observational studies (e.g. Koren et al. 2005; Sherwood 2002; Jiang et al. 2009) and modelling studies (e.g. Fan et al. 2013; Khain et al. 2005; Morrison and Grabowski 2011; Storer and van den Heever 2013) show it (see also IPCC AR5, Boucher et al. 2013). We have added these references to the revised manuscript on Section 1.

**Identification of deep convective clouds might further improve the results. MODISdetected cloud top pressure and CloudSat data would be helpful to select the cases with deep convective clouds.**

Also identification of deep convective clouds separately would be useful in a cloud-scale study with a lagrangean approach. Such an approach, however, would be different than in our study. We have used a precipitation limit to separate cases with deep convection to cases without precipitation or precipitation from shallow clouds. We have added the following explanation to the revised manuscript Section 2.4:

"We use the limit of 1 mm to exclude cases without any deep convection from the study."

**It should be noted and be mentioned in the article that using AOD as a measure of CCN concentration may introduce substantial uncertainties which is dependent on the aerosol type, vertical profile and hygroscopic growth.**

We agree with the Referee #1 that using AOD as a measure of CCN concentration may introduce uncertainties which depend on the aerosol type, vertical profile and hygroscopic growth. We have added the following sentence to section 2.1:

"However, it should be noted that AOD does not provide information on aerosol type, vertical profile of the aerosols or hygroscopic growth of the aerosol particles." Causality questions related to the hygroscopic growth of the particles are further discussed in section 3.2.

**The discussion on long-wave radiative effect is somewhat incomplete and unclear. For instance, the time period of the calculated the top-of-the-atmosphere radiative effect should be clarified, monthly mean or instantons values, in Section 2.5. Moreover, discussions on radiative effect are too short in Section 3.3. It would be better to discuss the radiative effect in detail, such as the difference between tropical moist (TM) and mid-latitude dry (MLD) conditions.**

We have added text about the radiative effect and modified chapter 3.3 as following:

"Outgoing long-wave radiation is sensitive to the upper tropospheric water vapour concentration (Held and Soden, 2000). Water vapour is a greenhouse gas that mainly affects climate by absorbing outgoing longwave radiation. To balance the decrease in outgoing long-wave radiation that results from increased UTH, an increase in surface and lower atmospheric temperatures is required that acts to increase the outgoing long-wave radiation.

To quantify these effects, radiative transfer calculations (see Section 2.5) were conducted for two reference soundings of tropical moist (TM) and mid-latitude dry (MLD) air. The results show that a 1 %RH increase in the UTH of moist tropical air causes a local positive radiative effect of 0.25 W m$^{-2}$ (0.39 W m$^{-2}$ for dry air, see Table 3). In dry air the impact is larger as the fractional increase in water vapour concentration for a fixed increase in RH is larger in dry air (Held and Soden, 2000).

The observed summertime increase of 2.2 ± 1.5 %RH in UTH (5.8 ± 1.4 %RH without removal of AOD dependence on cirrus fraction) due to aerosols corresponds to the radiative effect of +0.5 ± 0.4 W m$^{-2}$ (+1.4 ± 0.3 W m$^{-2}$, interpolated from Table 3) in tropical moist air. The real radiative effect by this phenomenon may be even larger than these values, since the effect of wet scavenging may partially obscure the microphysical effects of aerosols on UTH in our study region."

**Technical Corrections:**
**Caption of Figure 11: please check the labels of panels in the caption.**

Labels of panels in Figure 11 have been corrected.

**Page 6, Line 8: MSL should be defined here.**
We have replaced MSL with "mean sea level" on page 6, line 8.

**Figure 10: the abbreviations like omega and dir in this figure need to be specified in the caption.**

Specifications of the abbreviations "omega" and "dir" has been added to the caption of Figure 10.

References:

Fan *et al*.: Microphysical effects determine macrophysical response for aerosol impacts on deep convective clouds, *PNAS*, 110, 48, 2013.

Held and Soden: Water vapor feedback and global warming. *Annu. Rev. Energy Environ.,* 2000, 25, 441-475, 2000.

Jiang *et al.:* Aerosol-CO relationship and aerosol effect on ice cloud particle size: Analyses from Aura Microwave Limb Sounder and Aqua Moderate Resolution Imaging Spectroradiometer observations. *J Geophys. Res.,* 114, D20, 2009.

Khain *et al.:* Aerosol impact on the dynamics and microphysics of deep convective clouds. *Q. J. R. Meteorol. Soc.,* 131, 611, 2639-2663, 2005.

Koren *et al.:* Aerosol invigoration and restructuring of Atlantic convective clouds. *Geophys. Res. Lett.,* 32, 14, 2005

Morrison and Grabowski: Cloud-system resolving model simulations of aerosol indirect effects on tropical deep convection and its thermodynamic environment. *Atmos. Chem. Phys.,* 11, 20, 10503-10523, 2011

Sherwood, S.: Aerosols and Ice Particle Size in Tropical Cumulonimbus. *J. Clim.*, 15, 9, 1051-1063, 2002.

Storer and van den Heever: Microphysical processes evident in aerosol forcing of tropical deep convective clouds. *J. Atmos. Sci.,* 2013, 70, 2, 430-446, 2013.

---

## Author Comment (AC2) · 14 Oct 2016

Observational evidence for aerosols increasing upper tropospheric humidity

Reply to Anonymous Referee #2

Laura Riuttanen et al.

Atmospheric Chemistry and Physics Discussions 14.10.2016

**Comments by Referee #2:**

**This is an interesting paper that studied the relationships between aerosol loading and upper tropospheric humidity using satellite remote sensing data. The paper is very well written and easy to follow. The authors have comprehensively explored many elements of uncertainty, however I still have some questions on the analysis. I suggest the publication of the manuscript after addressing some comments as follows.**

Authors' response:

We thank the Anonymous Referee #2 for his/her comments. The reviewer raises some questions in the specific comments. Please find our response below.

**Specific comments:**
**1. During the summer season, there might be diurnal variation of precipitations. Since the UHR change could result from the convective transport of marine boundary layer air to the upper troposphere and AODs are measured 11 hours ahead of URH, differences in diurnal variability between precipitation (UHR) and AODs are likely to contribute to the seen relationships. Suggest test it.**

We agree that there might be a diurnal cycle in convection in the area. In our analysis we have used AOD data obtained at 10:30 a.m., UTH data at 9:30 p.m. and precipitation data from 8:30 a.m. to 8:30 a.m. local solar time. As the time of the observation is always the same (once per day), diurnal cycle cannot affect our results.

**2. The binning approach is not totally convincing for excluding the meteorological/ synoptic effects on AOD. In addition, the value ranges of those bins are not clear. I would think more logically one should compare the 10-m wind speed and surface humidity between high AOD and low AOD scenarios and test the statistical significance of 10-m wind speed differences and surface humidity differences between the high and low AOD scenarios. Not only that AODs are influenced by humidity but also it has significant contributions from large size natural sea salt aerosols (their emission rates are wind speed dependent).**

We agree that AOD depends on both low level wind speed and surface humidity. However, the effect of AOD on UTH is seen in every class of 10-m wind speed and 2-m relative humidity – except the bin with the largest surface relative humidity. Therefore, although AOD depends on those meteorological variables, AOD's effect on UTH cannot be due to, e.g., wind speed increasing AOD and UTH separately. If wind speed would affect AOD and UTH separately, and thereby cause a positive relation between AOD and UTH, then binning by wind speed would remove, or at least diminish remarkably, such a relation.

Referee #2 asked about the values of meteorological parameters of bins in Figure 10. We have now added a mean value of each meteorological parameter in the bins to Figure 10.

**3. Is it possible to only use clear sky AOD (with cirrus fraction = 0 or < 0.1), given that cirrus clouds lead to biases in AOD values? Does the results change when using only clear sky AODs?**

Using only clear-sky AOD would be problematic considering the nature of the phenomenon. As the effect of aerosols on UTH is expected to occur via sublimation of anvil cirrus clouds, it is expected that some cirrus clouds remain in the same 1 deg x 1 deg grid box. So the occurrence of cirrus clouds also reflects the occurrence of deep convection in the area and therefore we would not recommend studying only cases with cirrus fraction below 0.1.

However, the results for cirrus fraction smaller than 0.1 can be seen in Figure R1. UTH in general is lower, as expected, when cirrus fraction is low. Also error estimates are larger. However, with large UTH values, the effect can still be clearly seen in the data (Figure R1 b).

[Figure]

Figure R1. Same as Figure 6 in the manuscript, but only for values with cirrus fraction below 0.1. Number of data points was 4636.

**Minor comments:**
**Page 7, Lines 10 and 15: I disagree that bulk microphysics cannot capture important aerosol effects, and only spectral bin microphysics can capture the effect. Through improved coupling between aerosols and microphysics in conjunction with maybe some improvement in the bulk microphysics (including ice nucleation scheme), there is no reason that more detailed bulk microphysics cannot simulate those effects.**

We have modified the last paragraph of Section 4 in the revised manuscript as follows:

"Namely, as noted by Khain et al. (2015), the only feasible option in current general circulation models is to use bulk microphysics parameterization schemes. However, bulk microphysics schemes have trouble in producing aerosols' effect on cloud cover and cloud top height. As a result, the effect of aerosols on UTH is not correctly included in the current projections of climate change produced by general circulation models."